# Treatment of fatigue with physical activity and behavioural change support in vasculitis: study protocol for an open-label randomised controlled feasibility study

Lorraine Harper,[1] Matthew David Morgan,[1] Dimitrios Chanouzas,[1] Hollie K Caulfield,[2] Linda Coughlan,[3] Caroline Dean,[4] Kate Fletcher,[2] Fiona Cramp,[5] Sheila Greenfield,[6] Catherine A Hewitt,[7] Natalie J Ives,[7] Sue Jowett,[6] Amanda Daley[8]

For numbered affiliations see end of article.

**Correspondence to**
Professor Lorraine Harper;
l.harper@bham.ac.uk

## ABSTRACT

**Introduction** Fatigue is a major cause of morbidity, limiting quality of life, in patients with antineutrophil cytoplasmic antibody-associated vasculitis (AAV). The aetiology of fatigue is multifactorial; biological and psychosocial mediators, such as sleep deprivation, pain and anxiety and depression, are important and may be improved by increasing physical activity. Current self-management advice is based on expert opinion and is poorly adhered to. This study aims to investigate the feasibility of increasing physical activity using a programme of direct contact and telephone support, to provide patient education, encourage behaviour self-monitoring and the development of an individual change plan with defined goals and feedback to treat fatigue compared with standard of care to inform the design of a large randomised controlled trial to test the efficacy and cost effectiveness of this programme.

**Methods and analysis** Patients with AAV and significant levels of fatigue (patient self-report using multidimensional fatigue index score questionnaire ≥14) will be randomised in a 1:1 ratio to the physical activity programme supported by behavioural change techniques or standard of care. The intervention programme will consist of 8 visits of supervised activity sessions and 12 telephone support calls over 12 weeks with the aim of increasing physical activity to the level advised by government guidelines. Assessment visits will be performed at baseline, 12, 24 and 52 weeks. The study will assess the feasibility of recruitment, retention, the acceptability, adherence and safety of the intervention, and collect data on various assessment tools to inform the design of a large definitive trial. A nested qualitative study will explore patient experience of the trial through focus groups or interviews.

**Ethics and dissemination** All required ethical and regulatory approvals have been obtained. Findings will be disseminated through conference presentations, patient networks and academic publications.

**Trial registration number** ISRCTN11929227.

### Strengths and limitations of this study

► Fatigue is a major cause of poor quality of life in patients with antineutrophil cytoplasmic antibody-associated vasculitis (AAV) and has no evidence base to direct treatment.
► This study will provide the evidence required to design a large randomised controlled trial to address whether physical activity improves fatigue in patients with AAV.
► The study is limited by its size; as a feasibility study, it has a small sample size and will not provide evidence of efficacy but will provide estimates to calculate sample size for a larger study

## INTRODUCTION

Fatigue is a common symptom limiting quality of life in patients with a wide range of inflammatory and other chronic diseases including systemic vasculitis.[1] Ninety-two per cent of patients with antineutrophil cytoplasmic antibody (ANCA)-associated vasculitis (AAV) consider fatigue as the most important symptom affecting their well-being,[1 2] and it has been causally linked to reduced social participation, social withdrawal and unemployment.[3 4] Despite the significance of the problem, there are currently no recommended therapies specifically for fatigue in patients with AAV. Current self-management advice provided by the National Institute for Health and Care Excellence (https://cks.nice.org.uk/tirednessfatigue-in-adults#!scenariorecommendation:1) and Arthritis Research UK (ARUK) (https://www.arthritis-researchuk.org/arthritis-information/daily-life/fatigue.aspx) is based on expert opinion and pragmatic advice without any

BMJ

underpinning evidence base. This is therefore, a major area of unmet need and has been identified as a priority research area by patients and ARUK.

The aetiology of fatigue is multifactorial[5 6] and related to a number of interacting central (such as reduced motivation and increased perception of effort) and peripheral (impaired muscle or cardiovascular function) biological and psychosocial mediators including sleep deprivation, pain, depression, lack of physical activity and reduced cardiovascular fitness, similar to other chronic diseases.[7–10] Only 47% of patients with AAV participate in at least 1 hour of moderate or vigorous physical activity per week (unpublished audit of 100 patients), and patients are reluctant to increase physical activity often due to fatigue, or for fear of worsening their fatigue.[11] Both central and peripheral components of fatigue need to be addressed in any successful treatment intervention.

Physical activity interventions improve sleep, quality of life, anxiety and depression in patients with chronic diseases and have been shown to improve fatigue in the short term.[12] However, studies with long-term follow-up often report no sustained increase in physical activity once support is discontinued.[13] The addition of cognitive behavioural support (CBS), as commonly implemented in successful chronic disease self-management programmes,[14] to a physical activity intervention may result in improved self-efficacy and a sustained increase in physical activity once the formal programme is completed.[15–18]

Our programme, using direct contact and telephone support, will provide patient education, encourage behaviour self-monitoring and the development of an individual change plan with defined goals and feedback to increase physical activity. These interventions have not been investigated to treat fatigue in patients with systemic vasculitis. A pilot study of five patients with AAV with fatigue suggested physical activity, supported by behavioural intervention, was an acceptable approach for patients with AAV and fatigue.[19]

## AIMS AND OBJECTIVES

The aim of this study is to assess the feasibility and acceptability of undertaking a phase III randomised controlled trial (RCT) of physical activity with behaviour change support including device assisted self-monitoring and telephone support with cognitive behavioural strategies compared with standard of care in patients with AAV.

The specific objectives of the study are to:
► Assess different recruitment methods to the trial for example, where and how patients are identified and recruited.
► Assess recruitment and retention rates in the trial.
► Assess the acceptability of the intervention and the burden of trial participation on patients.
► Assess adherence to the trial intervention.

► Assess the safety of the intervention with respect to possible injury and deterioration in fatigue levels as a result of participating in a physical activity trial.
► Assess the effect of trial participation on activity levels in patients in the standard care group.
► Test procedures for the trial, including delivery of the standardised protocolled intervention and administration of trial questionnaires and other outcome measurement tools, and identify areas that may improve the design and implementation of the intervention.
► Collect data on the variability of the outcome measures to inform a power calculation for a definitive trial.
► Provide an estimate of the cost of implementing the intervention.
► Collect feedback from patients on their experience of participating in the study.

## METHODS
### Overview of study design
This is a single centre, open-label randomised controlled feasibility study with a nested qualitative component. Feasibility will be determined by assessing recruitment, adherence to the intervention and retention rates. Acceptability and patient experience will be assessed through patient-reported outcome (PRO) questionnaires, and an optional nested qualitative study consisting of either focus groups or semistructured telephone interviews if insufficient numbers can be recruited to form focus groups.

Fifty participants will be randomised (via a computer-generated programme) in a 1:1 ratio to the intervention or standard care for this population (figure 1 study flow). The randomisation will be minimised by age (<65 vs ≥65 years). Participants randomised to the intervention arm will complete eight supervised (one per week) and 12 telephone coaching sessions over 12 weeks. Outcomes will be assessed at baseline, 12, 24 and 52 weeks from the first exercise session for those in the intervention group and from randomisation for those in the standard care group.

### Sample size
This is a feasibility study, and therefore a formal sample size calculation has not been performed. The study is not designed or powered to detect a statistically significant difference in efficacy between the two treatment arms.

A pragmatic recruitment target of 50 participants has been chosen as we expect this number will be feasible within a single centre with a population of 250 patients and should be sufficient to provide estimates of the feasibility outcomes. A sample size of between 24 and 50 is recommended for pilot studies to estimate sample sizes for a full trial.[20]

### Patient involvement
The study was designed in consultation with a range of patients. A patient research partner (CD) is part of the trial management group (TMG) and helped to design the

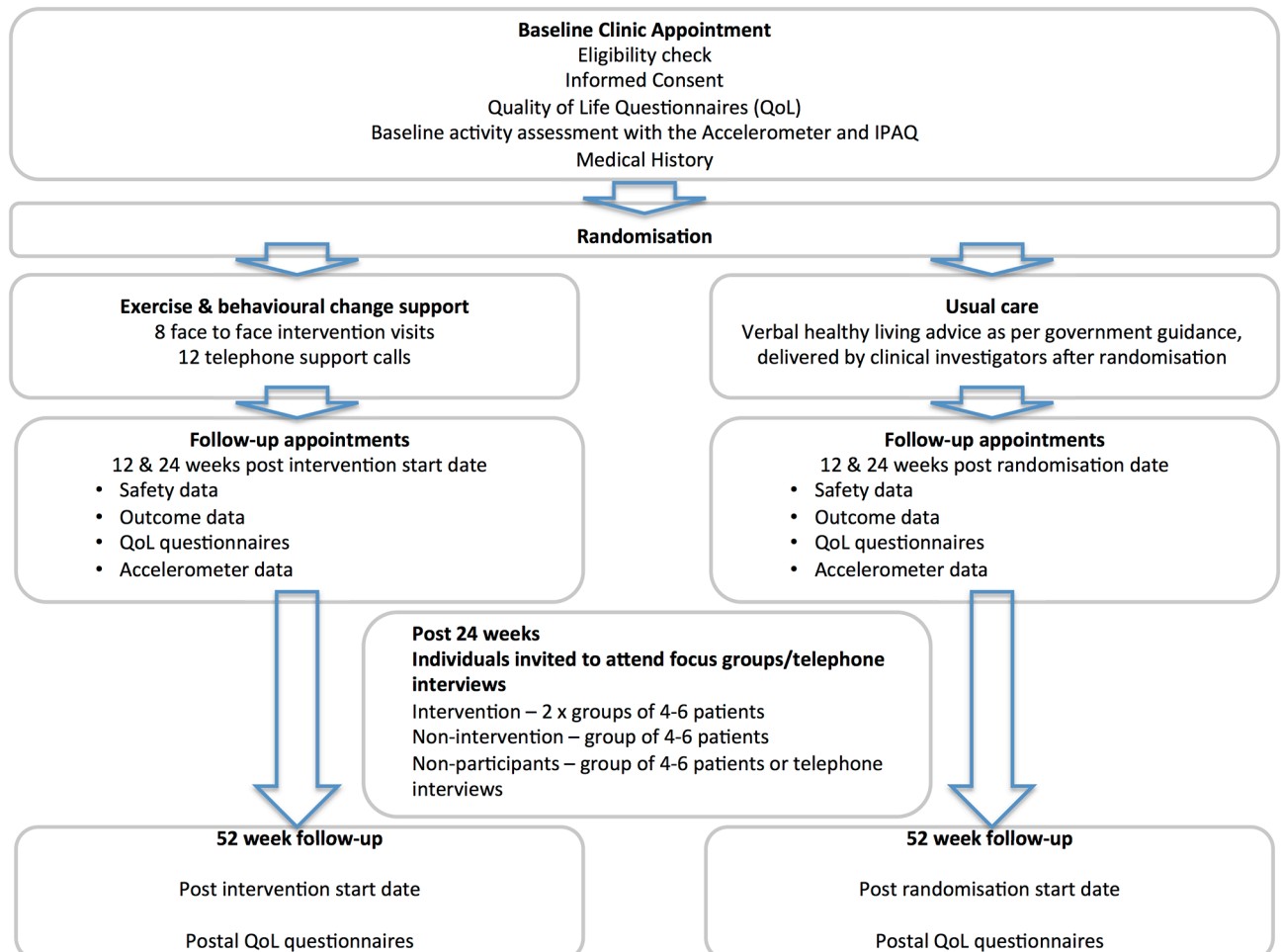

**Figure 1** Consort diagram.

study which was then discussed at patient support group meetings and with trustees of Vasculitis UK, the patient support charity. Feedback from these meetings was incorporated into the study design. Vasculitis UK has helped with patient recruitment and will help with dissemination of the study results.

### Participants and recruitment

Patients will be recruited to the study via the dedicated vasculitis service at University Hospitals Birmingham NHS Foundation Trust (UHB NHSFT) or via response to study adverts placed in the Vasculitis UK newsletter or other hospital clinics. Confirmation of eligibility for the study will be with the patient's own clinician. Patients are eligible if they are 18 years of age or older, have a diagnosis of AAV as classified by the European Medicines Agency Algorithm,[21] have been in remission for at least 6 months on the day of consent (defined by Birmingham Vasculitis Activity Score Version 3 (BVASv3)[22]=0 on day of consent and prednisolone dose <7.5 mg for 6 months) and have significant fatigue levels (multidimensional fatigue inventory (MFI)-20 general fatigue score ≥14). Exclusion criteria include inability to provide informed consent, inability or unwillingness to undertake physical activity, comorbidities considered by their own clinician

to contraindicate an increase in physical activity and an inability to understand and complete questionnaires.

### Standard care group

Patients randomised to the standard of care (control) group will receive their standard clinical care and will be provided with verbal activity advice as per UK government recommendations. Advice on activity will be given after randomisation, and patients will be advised to visit the government physical activity guidelines website.[23]

### Intervention group

The intervention is a physical activity and behavioural change support programme, plus standard care. The physical activity intervention will provide support for 12 weeks; weeks 1–8 will consist of weekly (where possible) direct contact in groups of 4–7 patients and additional individual telephone health coaching once weekly (where possible), then in weeks 9–12, telephone health coaching only will be provided. The intervention will be staged adapted, and participants will be encouraged to increase physical activity within their capabilities. Participants will be provided with personal activity self-monitoring devices, which are wrist-worn accelerometers (Fitbit Model FB405BKL), providing individual feedback

on daily physical activity. The supervised activity sessions will incorporate CBS (eg, goal setting, finding social support and understanding the costs/benefits of exercise etc) to promote long-term participation in physical activity. The intervention includes education, monitoring and assessment of progress and teaches skills to improve self-efficacy.

Individual specific, measurable, achievable, realistic and timed (SMART) goals[24] will be set by the participants with guidance from the intervention facilitator and will focus on increasing activity levels. The behavioural goal is for participants to achieve at least 30 min of moderate intensity activity, 5 days per week, ideally in bouts of 10 min, as per UK government recommendations, although any increase in physical activity or reduction in sedentary time will be viewed positively. The Fitbit activity self-monitor is supported with online web interfaces that facilitate effective components of physical activity interventions (https://www.fitbit.com/login).[25]

A detailed facilitator intervention manual has been developed to support the supervised sessions and telephone coaching. In brief, the physical activity intervention is designed to be pragmatic and accessible, taking into account exercise preferences and giving choices. During the direct contact sessions, participants will be asked to complete short bouts (eg, 5×3 min, with 2 min rest intervals) of low to moderate intensity aerobic exercise (eg, stepping ergometer, cycle ergometer, treadmill walking, rowing ergometer, arm cranking) at 50%–69% of predicted maximum heart rate (220—age) or 12–14 on the Borg Ratings of Perceived Exertion Scale.[26] Intensity will be monitored continuously during exercise training sessions. As the intervention progresses, participants will be encouraged to participate in longer periods of aerobic exercise (eg, 5×4 min) or to take shorter rests between bouts.

Where appropriate, participants will also be introduced to exercises for strength and control, which will typically involve two to six different resistance exercises (eg, wall press ups, arm curls, leg abduction, wall squats and/or regular squats, knee extensions, calf raises, sit-to-stand) each session. Body resistance, light weights and Therabands will be used to provide resistance and one to three sets of 5 to 20 repetitions will be performed, depending on level of disability and strength, as well as stage of the programme (exercises will be progressed according to individual capabilities and strength gains). Static stretching exercises for large skeletal muscle groups will be included in the sessions.

Participants will be encouraged to replicate the sessions at home supported by the telephone health coaching.

Using activity data collected from the Fitbit device from the preceding week, the intervention facilitator will assist participants to establish SMART goals.[24] Data collected will be shared with the therapist, via either the internet dashboard or paper diary. Review of daily and weekly physical activity will be undertaken to identify bouts of moderate/vigorous physical activity and to negotiate change plans to increase bouts of moderate/vigorous physical activity.

During the telephone discussion, participants will be encouraged to use their activity profiles generated from their Fitbit device, with data collected and stored using the online dashboard or paper diaries, to promote self-determination and self-regulation to achieve personal goals and maintain activity. The therapist will review fatigue, the impact of physical activity on fatigue and the management of fatigue with the participant. The therapist will review activity plans and understanding, provide physical activity education and help the patient to generate activity maintenance plans and set new goals.

### Study procedures

The study outcomes, measurement methods and assessments are summarised in table 1. Data are collected at baseline, and then at 12, 24 and 52 weeks, from the first exercise session for those in the intervention group and from randomisation for those in the standard care group. Data on recruitment, number of activity sessions and telephone coaching sessions completed and drop outs will also be recorded.

At the baseline, 12-week and 24-week assessments blood pressure, height (baseline only), weight and waist circumference will be measured, disease type, medical and drug history (baseline only), use of immunosuppressive therapy and assessment of damage associated with AAV, using the vasculitis damage index[27] (baseline only) will be taken. Laboratory investigations, including full blood count and kidney function are measured at baseline. At each assessment visit, disease activity will be assessed using the BVASv3. Clinical outcomes and safety of the intervention, including muscle or bone injury, disease relapse and cardiovascular adverse clinical events will be collected at 12 and 24 weeks as self-reported by the participant. All serious adverse events will also be recorded. Participants will complete a suite of validated health-related quality of life and fatigue questionnaires, which include short form-36 (SF-36),[28] ANCA Vasculitis questionnaire (AAV-PRO), EQ-5D,[29] MFI-20,[30] Pittsburgh Sleep Index,[31] Bristol Rheumatoid Arthritis Fatigue-Multidimensional questionnaire,[32] Hospital Anxiety and Depression score,[33] Brief Coping Orientation to Problems Experienced (Brief COPE) questionnaire[34] and the International Physical Activity Questionnaire long version at baseline and at 12, 24 and 52 weeks (at 52 weeks, the participants will be contacted by post and asked to complete the same questionnaires as completed at previous assessments and to return these by post or at next clinic visit). All participants will be asked to wear an accelerometer (GENEActive GATV01, Activinsights, Kimbolton, UK) for 7 days at baseline and at 12 and 24 weeks to provide an objective measure of their activity.

In order to undertake an analysis of all the direct resources and costs required to deliver the intervention, a detailed collection of health economic data will be embedded within the trial.

**Table 1** Visit overview

| Visit | Prebaseline | Baseline | Baseline (+4 weeks) | Weeks 1–8 | Weeks 9–12 | Week 12* | Week 24* | †Week 52* |
|---|---|---|---|---|---|---|---|---|
| Identification of eligible patients | x | | | | | | | |
| Eligibility check | x | x | | | | | | |
| Valid informed consent | | x | | | | | | |
| Relevant medical history taken | | x | | | | | | |
| Randomisation (without patient present) | | | x | | | | | |
| Activity advice telephone call (control arm only)‡ | | | | x | | | | |
| Exercise visits (intervention arm only) | | | | x | | | | |
| Weekly telephone coaching contacts (intervention arm only) | | | | x | x | | | |
| Follow-up clinic visits*§ | | | | | | x | x | |
| Final assessment (postal questionnaires)*§ | | | | | | | | x |
| Quality of life questionnaires | | x | | | | x | x | x |
| Focus groups or interviews (postintervention) | | | | | x | x | x | |

*The timing of the follow-up visits for intervention group are taken from intervention start date, not randomisation date; for the control group, the timing is taken from randomisation date.
†52 weeks follow-up assessment can be completed—2 weeks before due date and up to 8 weeks after the due date.
‡Control group have one telephone call in week 1 (postrandomisation) to receive standard activity advice only.
§12-week and 24-week follow-up assessments can be completed up to—2 weeks before due date and up to 3 weeks after the due date.

## Study outcomes and data analysis

The primary outcome of this feasibility trial is to determine if a full RCT comparing physical activity with behaviour change support versus standard care in patients with AAV is feasible. This decision will be made using a composite assessment of both quantitative and qualitative data based on a traffic light system using predefined stop-go criteria. The three criteria contributing to the traffic light system include[1] recruitment rates, defined as the proportion of eligible patients recruited into the study[2]; successful adherence to the intervention, defined as attendance at a minimum of 4 of the 8 visits for the face to face support, and acceptance of a minimum of 3 of the 4 telephone support calls in weeks 9–12 and[3] study drop-out, defined as complete withdrawal from the study, with no further data collected from the participant. The traffic light system comprises (1) green light: recruitment rate >50%; adherence rate >75% and drop-out rate <15% excluding those patients who drop-out or are unable to achieve adherence due to disease activity. If all three criteria are met, we will proceed to a full trial with the protocol unchanged (unless there is a clear message from the qualitative work that would improve the protocol). (2) Amber light: recruitment rate 30%–50%, adherence rate 50%–75% or drop-out rate 15%–30% excluding those patients who drop-out or are unable to achieve adherence due to disease activity. If one or more of our amber light criteria are met, we will adapt the protocol in light of the result of the feedback from the qualitative work and our experience to improve whichever criteria are not at the 'green-light' level before proceeding to full trial if possible. We will assess whether adaption of the protocol

will require a further feasibility study or pilot study before progressing. (3) Red light: recruitment rate <30%, adherence rate <50% or drop-out rate >30% excluding those patients who drop-out or are unable to achieve adherence for disease activity. If one or more of these criteria are met we would consider the current protocol not feasible and not progress to a full RCT.

The primary analysis to assess the stop-go criteria will be undertaken once all participants have completed the 24-week assessment, and corresponding outcome data has been entered onto the study database and validated as being ready for analysis. Feasibility outcomes including the three components of the stop-go criteria will be analysed by pooling the two randomised groups and presenting overall estimates with 95% CI. Reasons for non-entry into the trial will be assessed, particularly in relation to the patient eligibility criteria and reasons for patient refusal. Data on patients who stop the intervention and/or who do not complete the trial (such as withdrawals and those lost to follow-up) will be collected throughout the trial to allow assessment of patient adherence and retention rates. Reasons for non-adherence and non-completion will be analysed descriptively.

All clinical and patient-reported outcomes will primarily take the form of simple descriptive statistics (eg, proportions and percentages, means and SD) and where appropriate, point estimates of effects sizes (eg, mean differences and relative risks) and associated 95% CI. Data return rates at each time point will be assessed along with data completeness of the various outcomes measures. This will inform the use of appropriate questionnaires for the primary outcome in a future study. The

data will be analysed using the intention to treat principle, where patients are analysed in the treatment group to which they were randomised regardless of adherence to intervention or compliance with the protocol.

Detailed data collection of the resources required to deliver the intervention will allow a cost analysis to be undertaken to calculate the cost of implementing the intervention in a full RCT. The mean cost of the intervention per patient will be estimated both as per usage in the study and also the full cost assuming full compliance. Data collected directly from the trial will determine the resources required for delivering the supervised exercise and telephone health coaching. Resources will include staff costs, any equipment/consumables needed, printed material, telephone call costs, staff training costs and infrastructure (eg, room space). Information will be collected on number of patient contacts, length of time for face to face and telephone contacts and group size for the supervised exercise sessions. Standard unit costs for healthcare will be applied, with local costs sought from participating healthcare providers. Sensitivity analysis will estimate costs with changes to costing assumptions, for example, staff grade, group size.

### Nested qualitative study

Focus groups of 4–6 participants will be carried out to discuss their experience of the trial. Two groups will consist of people who participated in the trial and participated in the intervention, one group of people who participated and received standard of care and the final group will be made up of people who did not wish to participate in the trial. Individuals who did not wish to participate in the trial will be contacted and invited to participate in a focus group to better understand the reasons for non-participation. If we cannot form one or more of the focus groups described, then patients will be invited to participate in 1:1 semistructured telephone interviews. The qualitative phase will be conducted by a researcher not involved with the rest of the study and will collect data on the experiences of the study and the intervention, in more detail, and to gather suggestions for improvements to the design of a future RCT.

Focus groups and interviews will be recorded and transcribed verbatim. QSR NVivo 8 will be used for data management and data analysed using a framework approach,[35] facilitated by the use of to identify freely emerging themes and categories. Those who did not participate in the trial will be invited to complete the same questionnaires as collected during the trial. These will be analysed as above using simple descriptive statistics.

### Trial management and monitoring

The trial will be coordinated by the TMG (principal investigator, coinvestigators, trial coordinator, physical activity trainer and statistical advisors) in conjunction with the NIHR/WT CRF. A Trial Steering Committee (TSC) consisting of the principal investigator as well as three independent consultants, a statistician and patient advocate not involved in the study will provide the overall supervision of the trial. The TSC will oversee trial progress, protocol compliance, patient safety and review of updated information. As this is a feasibility study with short follow-up, no Data Monitoring Committee will be formed, as agreed with the sponsor. The integrity of data entry will be ensured using a trial-specific Data Input Quality Control standard operating procedure (Trial Master File, University of Birmingham). The trial database will contain date of birth and sex. Data analyses will be undertaken on pseudoanonymised datasets. All source data and original participant identities will be kept in a locked office in the trial site file only at UHB NHSFT or at Birmingham Clinical Trials Unit.

### Ethics

All patients will be provided with verbal and written study information. The study will be undertaken at a single site, and local governance approval has been granted by UHB NHSFT. Steps have been taken to ensure patient welfare when designing this study. The study complies with the International Conference for Harmonisation of Good Clinical Practice guidelines and the Research Governance Framework for Health and Social Care. Any protocol amendments will be submitted to the sponsor and relevant regulatory bodies for approval prior to implementation, and trial participants will be informed of any protocol modifications.

### Dissemination policy

It is anticipated that the findings of this study will be published in peer-reviewed journals, presented at national conferences, and that the results will be disseminated to all study participants who wish to be informed. It is anticipated that the results of this study will inform the design of a large RCT investigating the efficacy of increasing physical activity to treat fatigue in this population.

**Author affiliations**
[1]Institute of Clinical Sciences, College of Medical and Dental Sciences, University of Birmingham, Birmingham, UK
[2]Institute of Translational Medicine, College of Medical and Dental Sciences, University of Birmingham, Birmingham, UK
[3]NIHR/Wellcome Trust Clinical Research Facility, UHB NHS Foundation Trust, Birmingham, UK
[4]Patient Research Partner
[5]Faculty of Health and Applied Sciences, University of the West of England Bristol, Bristol, UK
[6]Institute of Applied Health Research, College of Medical and Dental Sciences, University of Birmingham, Birmingham, UK
[7]Birmingham Clinical Trials Unit, College of Medical and Dental Sciences, University of Birmingham, Birmingham, UK
[8]School of Sport, Exercise and Health Sciences, Loughborough University, Loughborough, UK

**Acknowledgements** Trustees and patient members of Vasculitis UK are thanked for their helpful contributions which informed the development of the protocol. The work was carried out at the National Institute for Health Research (NIHR)/Wellcome Trust Birmingham Clinical Research Facility. The views expressed are those of the author(s) and not necessarily those of the NHS, the NIHR of the Department of Health.

**Contributors** MDM, DC, HKC, LC, CD, KF, FC, SG, CAH, NJI, SJ, AD, LH: all authors developed and reviewed the protocol; conceived the data analysis plan and the writing of this article; critically reviewed and edited drafts and approved the final version of the manuscript. They also had full access to all of the data (including statistical reports and tables) in the study and can take responsibility for the integrity of the data and the accuracy of the data analysis. LH, the study guarantor.

**Funding** This work was supported by Arthritis Research UK grant number 21199. The study is sponsored by the University of Birmingham. The University of Birmingham holds public liability (negligent harm) and clinical trial (negligent harm) insurance policies, which apply to this trial. The sponsor has been involved in protocol design as well as the development of the case report form.

**Competing interests** None declared.

**Patient consent** Not required.

**Ethics approval** The protocol was reviewed and obtained a favourable opinion by the Health Research Authority (IRAS project ID 210364) on 29 September 2016 and adopted on to the NIHR portfolio.

**Provenance and peer review** Not commissioned; externally peer reviewed.

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
