## [Reviewer comments · BMJ Open]

ARTICLE DETAILS

TITLE (PROVISIONAL)	Treatment of fatigue with physical activity and behavioural change support in Vasculitis – study protocol for an open-label randomised controlled feasibility study.
AUTHORS	Harper, Lorraine; Morgan, Matthew; Chanouzas, Dimitrios; Caulfield, Hollie; Coughlan, Linda; Dean, Caroline; Fletcher, Kate; Cramp, Fiona; Greenfield, Sheila; Hewitt, Catherine; Ives, Natalie; Jowett, Sue; Daley, Amanda

VERSION 1 – REVIEW

REVIEWER	Thomas Wilkinson University of Leicester, UK
REVIEW RETURNED	30-Aug-2018

GENERAL COMMENTS	The current proposed study aims to provide feasibility evidence for a large RCT aimed to address whether physical activity improves fatigue in AAV patients. The protocol is well written and clearly set out, with detailed explanations of the rationale, methods, and analysis. I particularly found the use of a stop-go criteria to be well explained. Given the study has already received ethical approval and adopted to the NIHR portfolio, my (minor) comments only extend to the protocol submitted.  1) In the abstract, it would be nice to clarify on the MDF-index - is this a self-report questionnaire or one that is completed by the clinician? 2) Reference 10 in the introduction can be added to the sentence above as this statement appears to be a repetition on the opening lines in this paragraph 3) How will safety of the intervention be assessed? Self-report injuries or more severe adverse events? 4) In the first line of the methods, do you mean "...with a nested qualitative component" and not "focus component"? 5) Given the primary outcome is likely to be fatigue in a RCT - do you think having just 1 measure of fatigue is adequate? Why was this one chosen above others (e.g., FACIT, Chalder, etc.). 6) How was the stop-go progression decided? Did you use expert input/patient input or are they taken from elsewhere? 7) I would like to see a bit more detail on the economic costing - what exactly will you be recording? Do you mean things like costs of telephone calls etc.? Staff-hours taken to recruit? Other than these points, I think the study is well designed and wish you luck with it.
--

REVIEWER	Susanne Pettersson Karolinska Institutet, Sweden
REVIEW RETURNED	02-Sep-2018

GENERAL COMMENTS	According to the instructions the date of the study should be included I could not find such data. In the paragraph Interventions Group, you state an aim of increase self-efficacy, however I can not find any questionnaire to capture or evaluate this. You have multiple questionnaires to assess fatigue and HRQoL, I do not find any argumentation for this action that will only increase the questionnaire burden for the participants. The description of the Health economic calculations is sparsely, and should be explained further If this is a behavioral change study aiming to test the feasibility, analyses of the participants perceptions of the behavioral change techniques used in the study, you could for example look at the publications of Michie et al
---

VERSION 1 – AUTHOR RESPONSE

Reviewer 1

1. The abstract has now been modified to state “patient self-report using multi-dimensional fatigue index score questionnaire ≥ 14 ” as requested.
2. We have removed the repetitive sentence as requested.
3. Safety of the intervention will be assessed by self-report of injuries and cardiovascular events by the patients. The text has been amended to read “Clinical outcomes and safety of the intervention, including muscle or bone injury, disease relapse and cardiovascular adverse clinical events will be collected at 12 and 24 weeks as self-reported by the participant. All serious adverse events will also be recorded.”
4. We apologise for the typographical error the Reviewer is correct and the text has been amended to read “a nested *qualitative* component”
5. We agree with the Reviewer that it is important to use more than one method of collecting measures of fatigue in this population. We have included 3 measures of fatigue, SF36 which includes the vitality domain, the BRAF-MDQ and the MFI-20. All 3 of these questionnaires are completed at each assessment visit. The text currently reports that “participants will complete a suite of validated health related quality of life and fatigue questionnaires, which include Short Form-36 (SF-36)(28), ANCA Vasculitis questionnaire (AAV-PRO), EQ-5D(29), MFI-20(30), Pittsburgh Sleep Index(31), Bristol Rheumatoid Arthritis Fatigue- Multi-dimensional questionnaire (BRAF-MDQ)(32).....” We chose the MFI-20 as the inclusion criteria for the level of fatigue as we have previous experience of this questionnaire. There is no specific fatigue score for this patient population. As part of the feasibility study we plan to look at data completion and identify which of the fatigue measures performs best. This is now reported in the analysis section. “Data return rates at each time point will be assessed along with data completeness of the various outcomes measures. This will inform the use of appropriate questionnaires for the primary outcome in a future study.”

6. The stop-go criteria were determined by discussion within the steering group, which included a patient representative, informed by reports from other activity intervention studies. The criteria were supported by the independent grant review statistician for ARUK.
7. We have added additional detail on the economic costing which now reads “Detailed data collection of the resources required to deliver the intervention will allow a cost analysis to be undertaken to calculate the cost of implementing the intervention in a full RCT. The mean cost of the intervention per patient will be estimated both as per usage in the study and also the full cost assuming full compliance. Data collected directly from the trial will determine the resources required for delivering the supervised exercise and telephone health coaching. Resources will include staff costs, any equipment/consumables needed, printed material, telephone call costs, staff training costs and infrastructure (e.g. room space). Information will be collected on number of patient contacts, length of time for face to face and telephone contacts and group size for the supervised exercise sessions. Standard unit costs for healthcare will be applied, with local costs sought from participating healthcare providers. Sensitivity analysis will estimate costs with changes to costing assumptions, e.g. staff grade, group size.”

Reviewer 2

1. Thank you for pointing this out. We have included the date of trial registration on page 2.
2. The Reviewer rightly points out that we have not included any specific questionnaires to assess self-efficacy as this was not a specific aim of the study. The sentence has been amended to read “The intervention includes education, monitoring and assessment of progress and teaches skills to improve self-efficacy”.
3. We acknowledge that there are multiple questionnaires to assess quality of life and fatigue. Our patient feedback from our previous studies did not note that completion of these questionnaires was a burden for participants and the patient representative on our steering committee was comfortable with the level of questionnaire burden. There is no specific questionnaire to measure fatigue in this population. We wished to gain information as to which questionnaire performs best in terms of data completeness and this information will inform a future study.
4. The health economics section has been amended to increase the detail and now reads “Detailed data collection of the resources required to deliver the intervention will allow a cost analysis to be undertaken to calculate the cost of implementing the intervention in a full RCT. The mean cost of the intervention per patient will be estimated both as per usage in the study and also the full cost assuming full compliance. Data collected directly from the trial will determine the resources required for delivering the supervised exercise and telephone health coaching. Resources will include staff costs, any equipment/consumables needed, printed material, telephone call costs, staff training costs and infrastructure (e.g. room space). Information will be collected on number of patient contacts, length of time for face to face and telephone contacts and group size for the supervised exercise sessions. Standard unit costs for healthcare will be applied, with local costs sought from participating healthcare providers. Sensitivity analysis will estimate costs with changes to costing assumptions, e.g. staff grade, group size.”

5. We thank the Reviewer for their helpful sign-posting.

VERSION 2 – REVIEW

REVIEWER	Thomas Wilkinson University of Leicester, UK
REVIEW RETURNED	19-Sep-2018

GENERAL COMMENTS	The authors have addressed all my comments thoughtfully and descriptively. Thank you and I look forward to reading about the study upon completion.
---